# A Noninvasive Method for Time-Lapse Imaging of Microbial Interactions and Colony Dynamics

Carlos Molina-Santiago,[a,b] John R. Pearson,[c] María Victoria Berlanga-Clavero,[a,b] Alicia Isabel Pérez-Lorente,[a,b] Antonio de Vicente,[a,b] Diego Romero[a,b]

[a]Instituto de Hortofruticultura Subtropical y Mediterránea "La Mayora", Universidad de Málaga-Consejo Superior de Investigaciones Científicas (IHSM-UMA-CSIC), Málaga, Spain
[b]Departamento de Microbiología, Universidad de Málaga, Málaga, Spain
[c]Nano-Imaging Unit, Andalusian Centre for Nanomedicine and Biotechnology, BIONAND, Málaga, Spain

Carlos Molina-Santiago and John R. Pearson contributed equally to this work. Author order was decided according to total contribution in the manuscript.

**ABSTRACT** Complex interactions between microbial populations can greatly affect the overall properties of a microbial community, sometimes leading to cooperation and mutually beneficial coexistence, or competition and the death or displacement of organisms or subpopulations. Interactions between different biofilm populations are highly relevant in diverse scientific areas, from antimicrobial resistance to microbial ecology. The utilization of modern microscopic techniques has provided a new and interesting insight into how bacteria interact at the cellular level to form and maintain microbial biofilms. However, our ability to follow complex intraspecies and interspecies interactions *in vivo* at the microscopic level has remained somewhat limited. Here, we detailed BacLive, a novel noninvasive method for tracking bacterial growth and biofilm dynamics using high-resolution fluorescence microscopy and an associated ImageJ processing macro (https://github.com/BacLive) for easier data handling and image analysis. Finally, we provided examples of how BacLive can be used in the analysis of complex bacterial communities.

**IMPORTANCE** Communication and interactions between single cells are continuously defining the structure and composition of microbial communities temporally and spatially. Methods routinely used to study these communities at the cellular level rely on sample manipulation which makes microscopic time-lapse experiments impossible. BacLive was conceived as a method for the noninvasive study of the formation and development of bacterial communities, such as biofilms, and the formation dynamics of specialized subpopulations in time-lapse experiments at a colony level. In addition, we developed a tool to simplify the processing and analysis of the data generated by this method.

**KEYWORDS** biofilms, microbe-microbe interactions, microbial communities

Bacteria commonly live in structurally and dynamically complex biological systems forming surface-associated communities known as biofilms, which typically feature one or more bacterial species and/or other organisms nearby (1). Complex interactions at nutritional and chemical levels can confer a range of benefits, including increased resistance to environmental stresses, metabolic exchange, better surface colonization, and increased horizontal gene transfer, which can greatly affect the final structure of the entire community (2, 3). Thus, cooperative or competitive interactions can lead to mutually beneficial coexistence or the displacement or death of particular organisms or subpopulations (4). In addition to single and multispecies biofilms, bacterial interactions between different biofilm populations are of interest for a wide variety of

Address correspondence to Diego Romero, diego_romero@uma.es.

The authors declare no conflict of interest.

**TABLE 1** Comparison between 2D and 3D approaches used to visualize bacterial growth and their interactions

| Method | Monitoring time | Resolution | Invasiveness | Continuity | Reference |
|---|---|---|---|---|---|
| 2D imaging with standard camera in incubators | Up to 4–7 days | Centimeter to millimeter scale | No | No | Traxler et al. (13) |
| 2D imaging with standard camera in MOCHA | Up to 40 days (without contamination) | Centimeter to millimeter scale | No | Yes | Penil-Cobo et al. (10) |
| 3D imaging by cryostat sectioning | Up to 4–7 days | Micrometer scale | Yes | No | Vlamakis et al. (20) |
| 2D Time-lapse confocal microscopy | Up to 1 wk | Micrometer scale | No | Yes | Nadezhdin et al. (35) |
| 3D Time-lapse confocal microscopy | Up to 1 wk | Micrometer scale | No | Yes | Molina-Santiago et al. (14) |

applications such as the study of antibiotic resistance mechanisms and the development of new antimicrobial drugs (5–7). In these cases, bacterial populations are separated in the range of centimeters to millimeters, allowing long-distance effects to be studied.

Visualizing bacterial growth has been at the core of microbial science for over a century. Since the invention of the microscope in the 17th century, classical brightfield microscopy has long been used for the detection and classification of bacteria. Most of our understanding of bacterial interactions has been obtained through traditional microbiological techniques, which mainly address macroscopic changes visible to the naked eye (8). In recent years, light microscopy has been used to study whole bacterial populations, aiming to understand bacterial interactions and/or biofilm formation dynamics at the cellular and subcellular levels (9). The development of bacterial communities was initially documented using standard digital cameras that permit the capture of macroscopic 2D images of colonies grown on solid culture media (8, 10, 11). Changes in colony attributes were followed over time by placing inoculated plates in incubators for several days and imaging at different time points during the experiment (12, 13). This simple methodology is ideal for experiments where only limited changes occur over time, limiting dehydration and the potential problems caused by condensation because plates remain in optimal growth conditions for most of the experiment. However, these methods are limited in the study of more complex biological processes or microbial interactions. More sophisticated imaging approaches have been developed to deal with these issues. The MOCHA (microbial chamber) method (10), also based on macroscopic imaging, consists of a sealed chamber with built-in humidity control, and crucially, houses the camera inside the chamber, and was all controlled by an external computer (Table 1). With this method, petri dishes are prepared with paper wicks connected to a water reservoir to maintain hydration of the culture medium. Altogether, these improvements allow for the imaging of bacterial growth for long periods (up to 40 days) while monitoring environmental conditions inside the chamber. A similar approach was developed by Zacharia et al. (11) to perform time-lapse imaging using a fluorescence microscope and a USB-controlled heated mug warmer with a temperature sensor to monitor real-time temperature within the chamber and prevent condensation.

All the above imaging methods are macroscopic and lack the resolution to study bacterial interactions at the cellular level. The classical method based on mounting bacterial cultures between slides and coverslips is excellent for counting bacteria, visualizing cellular morphology, and other bacterial properties (14–16), but it is not useful for colony progression studies given the loss of all spatial and structural information. In recent decades, the development of new microscopes and the increasing importance of microbial ecology and biotechnology have led to the establishment of a variety of strategies for the study of microbial interactions and community development at the microscopic scale. For example, different lens objectives (from 4× to 10×) have been used to obtain time-lapses of entire colonies growing as biofilms (4, 17, 18). This has been complemented through the development of bioinformatics tools for analysis of the images obtained, such as BiofilmQ (19). In addition, some strategies have used CLSM to perform a detailed analysis of microbial interactions and biofilm developments in 3D.

For example, cryostat sectioning has been used with fluorescently labeled bacteria to demonstrate the existence of heterogeneous populations dedicated to different physiological functions and follow changes during *B. subtilis* biofilm development (20) (Table 1). However, this is an invasive and destructive method. Its temporal resolution was limited to the number of colonies frozen and sectioned, making it difficult to link sequential events that may vary between individual colonies. Another approach consists in combining high-resolution fluorescence microscopy with bacteria expressing different fluorescent reporters to characterize spatial expression patterns in colonies of *Bacillus subtilis* (21). This approach consists of cutting a piece of agar containing the colony edge and flipping it onto a glass-bottomed dish before imaging, thus allowing visualization of the reporter construct expression patterns at different levels of the bacterial colony (Table 1). Granato and Foster (22) have also used confocal microscopy to analyze the evolution of bacterial populations using air objectives during short-time experiments.

To study interactions between bacterial biofilm populations on solid media, we developed a method that combines microscopic resolution, 3D acquisitions, differential labeling, and continuous imaging for up to 5 days (14). This approach takes advantage of high-quality long working distance immersion optics combined with glass-bottomed petri dishes to visualize cellular dynamics within growing colonies, in a truly noninvasive manner (Table 1; Fig. 1A). Here, we described the acquisition methodology step-by-step, explaining the rationale behind it and its main advantages and disadvantages. We also detailed our post-acquisition methodology, which includes a freely available data handling tool (https://github.com/BacLive/) that aids in efficient processing and analysis of the results generated by the BacLive method. Finally, we showed specific examples of how this methodology could be used in (i) the study of bacterial biofilm interactions on solid media, and (ii) dynamics of the formation of specialized subpopulations within a *B. subtilis* biofilm.

## RESULTS AND DISCUSSION

**Acquisition methodology.** This article describes a novel method for noninvasive high-resolution imaging of bacterial colonies and biofilms, here termed BacLive. The BacLive method can be broadly divided into two sections: (i) image acquisition and (ii) data-handling/image processing. First, we provided a detailed overview of the most important acquisition variables to help other researchers best adapt the flexible BacLive method for their applications. Second, we described the associated BacLive ImageJ script, which aids data handling for subsequent image processing and analysis. Finally, we provided examples of how BacLive can be used for the noninvasive study of bacterial colonies and biofilms. The configuration we used for the BacLive method is summarized in Fig. 1B with a complete summary provided in the Materials and Methods.

The core of the BacLive acquisition methodology was the discovery that high-resolution microscopy imaging could be achieved while traversing solid agar growth media at thicknesses that allow bacterial colony growth and interactions to proceed normally. This was relevant because it permitted the analysis of bacterial colonies (bacterial interactions, biofilm development, etc.) without sample manipulation. This was enabled by using high numerical aperture immersion objectives with sufficient free working distances to focus through to the nascent colonies. Although the solid culture medium was relatively dense, most were almost transparent and homogeneous. Indeed, we observed little or no attenuation of fluorescence signal through the ∼1.3 mm of MSgg or LB solid medium and could readily resolve individual bacteria expressing fluorescent reporter constructs (Fig. S1A). This is because transparency and imaging depth is primarily determined by variations in refractive index within a medium (23). In contrast, significant fluorescence signal attenuation occurred in a relatively short distance upon excitation light crossing into the bacterial colony itself (Fig. S1A). This was due to colonies being composed of complex mixtures of proteins, water, and lipids, making it

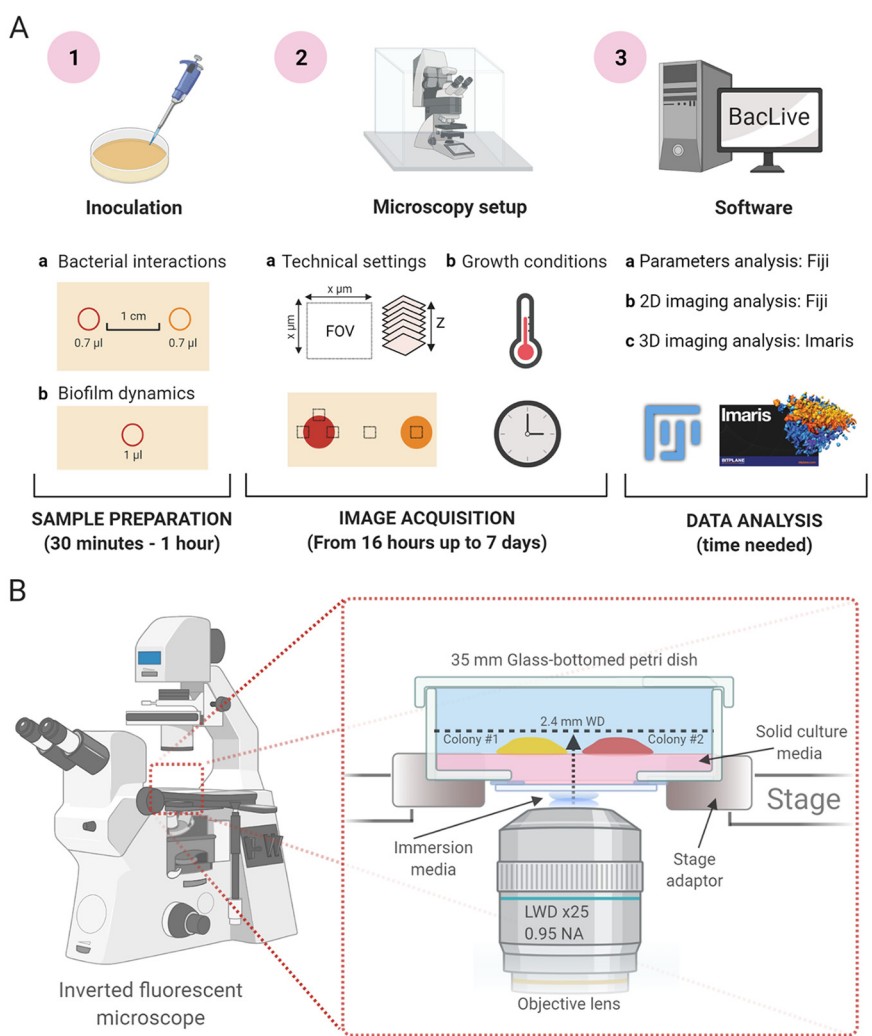

**FIG 1** Schematic representations of BacLive and plate position in an inverted fluorescence microscope. (A) Schematic representation of the main steps to perform bacterial interactions and biofilm development assays using confocal microscopy and Bac-Live processing image. Step 1 highlights the most important points for sample preparation. Step 2 indicates the main concerns related to microscopy setup and conditions, including multiposition acquisition, growth conditions, and technical setting as the number of slices in a Z-stack and image dimension. Step 3 focuses on data analysis using Bac-Live as the main tool for data image processing using FiJi and more complex analyses using commercial software such as Imaris. (B) Schematic representation of the plate position in an inverted fluorescence microscope which are highlighted the main elements that need to be considered for a correct setup, such as the need for immersion media, lens objective characteristics, the position of colonies for bacterial interaction, and the use of 35 mm glass-bottomed petri dishes.

more difficult for excitation and emission light to pass through it, as was the case for most complex biological samples (see below).

In theory, the BacLive method could be combined with any fluorescence microscopy modality. However, we obtained the best results using confocal laser-scanning microscopy (CLSM). CLSM has emerged as one of the dominant methodologies for fluorescence bioimaging over the last 30 years (reviewed in reference [24]). In the context of biofilm imaging, a confocal system provides three main advantages over widefield ones. First, they are more robust at optically reducing extraneous fluorescence, agar auto-fluorescence, and reflections, second, they provide higher-resolution 3D information of colony and biofilm development, and third, they are extremely flexible allowing different imaging conditions, resolutions, and magnifications to be specified depending on experimental requirements (see below). On the other hand, CLSM systems have some disadvantages for bacterial imaging. First, they tend to be relatively insensitive

compared to widefield systems, largely due to their use of less efficient detectors than the specialist cameras used with comparable widefield fluorescence microscopes (24). Moreover, the laser power used in CLSM systems is typically associated with higher photobleaching and phototoxicity. However, on balance, we found the better ability of CLSM to deal with background fluorescence, reflections, and bacterial autofluorescence was more significant than pure sensitivity for biofilm imaging. Even performing long timelapse studies using 10× and 25× objectives, we did not observe significant photobleaching, although the risk of photobleaching and phototoxicity will increase at higher magnification levels.

One of the key innovations in our methodology was the idea of imaging colonies through the agar substrate. Crucially, it allows immersion objectives to be used to visualize biofilm dynamics with a greatly improved resolution versus nonimmersion objectives. It also means that imaging can be performed on a closed petri dish without a loss of imaging quality (Fig. S1B and C). The crucial limiting factor was the working distance of the lens objective. Most high-quality immersion lens objectives were only capable of focusing up to 90 to 200 $\mu$m into a sample, which would make it impossible to focus through a reasonable volume of a solid substrate. An increasing number of high-quality, long working distance (LWD) objectives are being offered by microscope manufacturers. For our work, we took advantage of a Leica 25× water immersion objective (HC FLUOTAR L 25×/0.95 W VISIR) with a free working distance of 2.4 mm. The objective had a relatively high numerical aperture (NA) meaning a theoretical maximum XY resolution of 205 nm, which was more than sufficient for individual bacterial imaging in a confocal setup, depending on scan conditions. The free working distance of the objective determines how much solid medium could be used while still allowing us to focus on the nascent colony at the start of the experiment. We tested the ability of different bacterial strains to grow on different quantities of Lysogeny-Broth (LB) and Msgg solid medium and found normal growth occurred with as little as 1.3 mL of solid medium in a 35 mm diameter petri dish with a starting thickness of 1 mm while obtaining good quality images in both solid media. However, normal growth was affected if volumes were reduced below 1.3 mL (Fig. S1D).

**Confocal imaging flexibility and trade-offs.** CLSM systems enable both imaging size in pixels and the field of view (FOV or zoom) to be flexibly defined. In practice, this means it is possible to choose a larger FOV and smaller image sizes (e.g., 512 × 512 pixels) where we want to observe larger colony areas, allowing multiple imaging positions, higher temporal frequency, and/or longer timelapse duration at the cost of lower image quality and resolution because the effective pixel size was much larger than the available optical resolution. On the other hand, using the same lens objective we could choose a 1024 × 1024 pixel image format and decrease the FOV (i.e., increase the zoom) to obtain pixel sizes closer to the real optical resolution of the lens objective. This generates a much higher image quality at the cost of larger data sizes which may not allow as many imaging positions or require shorter experiments. The effect of these tradeoffs when using the HC FLUOTAR L 25×/0.95 W VISIR long working distance water immersion objective is shown in Fig. S1B and C. In our laboratory, we were more interested in the study of bacterial populations than individual bacteria. Thus, we routinely acquired up to 6 different positions with an image resolution of 512 × 512 pixels over 4 to 5 days to balance temporal resolution, data size, and image quality (see examples below).

**Long-term imaging with immersion lens objectives.** Evaporation of the water immersion medium is a major problem for long-term time-lapse imaging. Typically, after about 2 h, the experiment would have to be discontinued to add new water immersion media, which is impractical in multiday time-lapse imaging. For many microscope/objective combinations, automatic water dispensers are available that replenish water at the lens objective throughout the experiment. Because this kind of dispenser was not available in our case, we took an alternative approach, using a special silicone oil-based immersion medium (Immersol W 2010 Immersion Oil, Carl Zeiss), with a refractive index of 1.33 suitable for water immersion lens objectives. We found that this

method worked very well with our Leica lens objective for this kind of long-term experiment. For multiposition experiments (see below), the best results were obtained when oil was spread across the glass bottom, in addition to oil being added directly to the objective. By doing so, we were routinely able to image bacterial dynamics, uninterruptedly, over multiple days.

Most research microscope lens objectives are optimized for an ideal coverslip thickness of 170 $\mu$m. Focusing on different thicknesses of glass or materials with different properties can reduce resolution, and sensitivity and cause other optical artifacts. Therefore, for optimal results, we used 170 $\pm$ 10 $\mu$m coverglass-bottomed petri dishes (Fig. 1B). It is worth noting that some objectives have correction collars that can compensate for different coverslip thicknesses and that low magnification nonimmersion objectives will normally be less sensitive to suboptimal optical conditions and may allow visualization through standard plastic-bottomed petri dishes. However, given the already challenging optical path traversing the agar, we would strongly recommend avoiding this option if possible.

**Methods for dealing with evaporation and media shrinkage.** Our methodology allows for high-quality imaging while maintaining the petri dish closed, which helps minimize evaporation. However, especially in a heated environment, evaporation is inevitable. Evaporation changes the relative position of the colony with respect to the objective lens, so the colony would gradually drift out of focus after a short period (Fig. 2A). Evaporation could be reduced using auxiliary systems similar to those employed by the MOCHA method Penil Cobo et al. (10) where a lower petri dish that has a water reservoir is used to feed the upper agar petri dish with water through a paper wick where the bacteria are growing. Alternatively, an autofocus method based on fluorescence intensity or contrast could be employed to follow colony position changes. For our methodology, we chose to use an approach based on the acquisition of large 3D volume(s) that cover the future positions of the colony. We discovered that colony movement due to evaporation and medium consumption was highly consistent and predictable, with a constant movement over time of approximately 2.5 $\mu$m per hour (Fig. 2B) with negligible differences observed between different regions of the same plate (Fig. 2B zoom). Thus, it was possible to calculate the total volume needed to follow colony dynamics over the planned length of the experiment (Fig. S2A). This "brute force" approach meant that we were not dependent on specific fluorescence expressions to ensure correct autofocusing. Furthermore, it also ensures that we could follow biofilm dynamics in 3D without having to know in advance how colony development would proceed. The main limitation to this approach comes from data management and system stability. Data volumes can be reduced by taking advantage of the predictability of colony movement and performing acquisition in sub-volumes to cover different periods as illustrated in Fig. S2B Many microscope systems allow this kind of more complex acquisition pattern to be programmed automatically but could also be implemented manually every 24 h, for example. In practice, even with a 10-year-old Leica SP5 CLSM system, we could routinely acquire several plate positions over multiple days generating very large Z-series with final data volumes exceeding 20 to 30 gigabytes. However, this approach does bring several considerations and data processing issues as we discuss below.

**Other experimental considerations.** Research microscopes are arranged in either upright or inverted configurations, where the objective lens is located above or below the sample, respectively. Fig. 1B shows how bacterial growth could be followed in the inverted configuration, which has been successfully used in our experiments, but upright systems should work well and may have some advantages. Using a whole microscope heated enclosure, we did not have significant problems with condensation build-up. However, this potential problem would be reduced using an upright system, where the culture medium is inverted, and excess condensation would not drip onto the agar surface.

For most bacteria, temperature control is a crucial factor in colony growth, biofilm formation, and their interactions with other organisms. Therefore, it is crucial to control

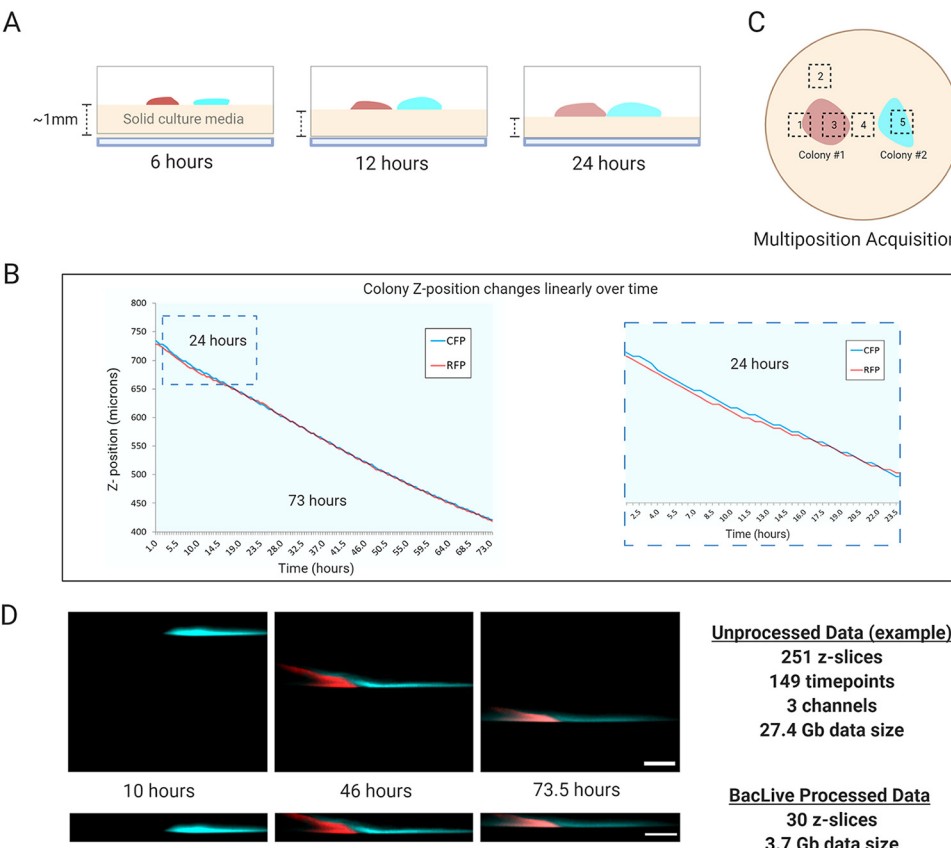

**FIG 2** Technical settings for experimental processing and data analysis. (A) Development of solid media thickness along with the experiment. Due to evaporation and bacterial consumption, the solid media shrinks over time. (B) Colony Z-position changes occurring in a timelapse (72 h) pairwise interactions experiment (strains labeled with CFP and RFP). Zoom for the first 24 h shows slight differences between fluorescent strains but, overall, the evaporation is constant throughout the whole experiment making it possible to predict axial colony movement and compensate for it. (C) Schematic representation of a multiposition experiment. Each square represents one image-capture position. We usually selected edge positions, inner colony positions, and expected contact positions. (D) A comparison of unprocessed and post-BacLive processed image data shows the differences according to the number of slices and data size reduction. The number of slices is dramatically reduced due to the filtering and selection of fluorescent slices.

the environmental conditions as much as possible (Fig. 1A). For long-term imaging, maintaining temperature is also critical for optical stability. When possible, we recommend the use of a full microscope enclosure with temperature control and a temperature sensor positioned as close as possible to the petri dish. Thus, variations that could affect bacterial growth or optical stability are reduced. Enclosure-based heating should also reduce condensation problems compared to a plate/stage heating solution.

Conditions will vary greatly depending on the species of bacteria being studied and the experimental goals. In most cases, experimental temperatures will range from 28°C to 37°C. In practice, temperature control is easier the greater the difference from the ambient temperature. Both the microscope excitation laser (or fluorescence lamp) and microscope electronics will contribute to a locally increased temperature close to the sample. In experiments with an ambient temperature of 21°C, we routinely measured temperatures adjacent to the sample of 26 to 27°C during long time-lapse experiments without any additional heating. Therefore, it would be difficult to perform experiments at lower temperatures without lowering the ambient temperature or active cooling. This emphasizes the importance of regulating temperature based on measurements that are as close to the sample as possible.

**Acquisition frequency/imaging conditions.** For a 3D time-lapse experiment, the distance between acquisition slices, or Z-sampling frequency, determines the 3D

resolution and has a critical effect on data size and acquisition time. Thus, it is often necessary to sacrifice 3D resolution to allow for a reasonable acquisition speed. We would typically set the Z-sampling interval between 1 and 3 $\mu$m depending on the experiment. Given that the axial resolution of our objective is ~550 nm, our data sets are considerably under-sampled, optically speaking, but are still sufficient to provide valuable 3D information about colony dynamics while limiting acquisition time and data set size.

Imaging conditions are another major consideration where image quality, sensitivity, laser power, and imaging speed must be balanced. It will normally be necessary to perform pilot experiments to establish appropriate conditions as the expression of fluorescent proteins can vary greatly between different constructs, strains, and at different stages of colony growth. In the example described in more detail below, each multichannel 2D image or Z-slice had an acquisition time of ~3 s. For a 40-h time-lapse experiment, we predicted a Z drift of ~100 $\mu$m but choose to acquire a larger 250 $\mu$m volume in case we wished to extend the experiment beyond the initial 40 h. We combined this with an ~2 $\mu$m Z-stack sampling frequency, equating to a total of 128 slices and ~6-min acquisition time per time point. We established a 30-minute interval as being sufficient to follow bacterial growth and biofilm dynamics, thus capturing 84-time points and a total of 32,256 images (2 fluorescence channels and transmitted light).

**Data set size, multipoint acquisition, and temporal resolution.** For our studies of biofilm dynamics in the interaction between *Pectobacterium carotovorum* and *Bacillus amyloliquefaciens* FZB42 (see 'study of bacterial population dynamics in interactions on solid media' section below), we wanted to be able to follow changes in each colony and their interaction during the same experiment. To do this, we took advantage of the motorized stage and multipoint acquisition, a feature common to many mid-to-high-end research microscopes. At the start of the experiment, we selected stage positions corresponding to different colony regions and regions where we predicted colonies might come into contact. We then defined Z-stack positions for each region separately because the agar surface is usually not exactly level with respect to the microscope, but otherwise defined them using the same total volume and Z-sampling frequency (Fig. 2C). Multiposition time-lapse acquisition proceeds by acquiring each Z-stack in turn. The time-lapse interval defines the period between starting the first Z-stack of each time point, thus all acquisitions must be completed within the time-lapse interval. This ensures that all fields have the same imaging interval even though they are not simultaneous. Acquiring multiple microscope fields also multiplies all of the limiting factors in terms of balancing image quality and imaging time. Reducing image resolution from 1024 × 1024 to 512 × 512 was typically necessary to avoid excessively long time points and, especially, to avoid generating prohibitive amounts of data.

As we observed, many variables need to be balanced in terms of bacterial inoculation, culture media, and acquisition settings to optimize a time-lapse experiment. Different bacterial models and experimental priorities require different compromises to be made for practical reasons. However, as with any time-lapse experiment, it is essential to validate the imaging configuration with appropriate controls to monitor the extent that imaging conditions might alter bacterial and biofilm development. If incubator-grown control plates show different patterns of growth or interaction then it may be necessary to further optimize environmental conditions or alter imaging conditions (e.g., reduced laser power). In some cases, the development of improved reporter constructs with enhanced expression levels or better stability may help achieve better results.

**Image processing methodology. (i) Data management (in general)/data compression *in situ*.** The primary disadvantage and limiting factor of the strategy we implemented in this method is the very large amounts of data that are typically generated. Fig. 2D shows an example where a 73-h acquisition generated 27.4 Gb of data per position. We routinely used this method to acquire data from 6 different positions over 5 days. Given a typical plate z-drift of ~2.5 $\mu$m per hour, this requires a minimum of ~300 $\mu$m total Z-stack thickness and 150 image sections given a 2 $\mu$m z-spacing. In the example shown below, we acquired two fluorescence channels for CFP and GFP

plus brightfield. Using a 512 × 512 XY resolution and 8-bit images this is equivalent to 0.66 Gb per time point or ~158 Gb over 5 days with a 30-minute acquisition interval. Therefore, finding efficient ways to deal with these large data sets is the focus of the remainder of this work.

**(ii) BacLive data preprocessing ImageJ/FIJI macro.** By acquiring large Z-stacks that cover the predicted trajectory of colony growth, media consumption, and media shrinkage, most of the image slices collected will inevitably contain non-useful data. Our objective is to trim this excess data while compensating for colony movement over time, thus allowing us to follow biofilm dynamics and colony interactions. Our strategy to do so was based on the observation that under our acquisition conditions, colony movement is linear over time (Fig. 2B). Therefore, by knowing the agar surface starting and ending positions, we could calculate the average change in Z position over time in terms of Z-slices. Using this value, we aimed to process the full data set using a "floating window" crop of Z-slices that follows the agar surface/colony position (Fig. 2D).

This relatively simple strategy is surprisingly effective in terms of compensating for colony movement/media shrinkage. However, it is still very time-consuming when performed manually for each time point. To make this process more efficient we devised a macro script written for the ImageJ/FIJI open-source image processing packages (25, 26). The script (freely downloadable from https://github.com/BacLive/) assists the user in detecting the average Z-slice change value and then applying a floating window crop and generating processed 3D time-lapse data set, where the plate surface should remain relatively fixed in the Z-plane and eliminating most of the unneeded Z-slices. Because ImageJ/FIJI supports a wide range of image formats through the Bioformats plugin (27), the same methodology should be effective regardless of the microscope type used for acquisition.

Over time our BacLive processing macro has developed considerably, offering a relatively straightforward interface for image processing, that is designed to avoid some of the memory limitations that can be problematic when dealing with large time-lapse data sets. For example, it is now compatible with virtual data sets opened with the Bioformats plug-in, making it relatively painless to process huge data sets without needing specialist workstations with large amounts of RAM. Processing is essentially divided into two sections as shown in the BacLive workflow (see Fig. S3). First, it gives the user the option of generating a graph showing average image intensity by channel, Z-position, and time. This will aid in choosing appropriate time points and channels for subsequent steps for tracking plate surface movement. Next, BacLive calculates the rate of surface movement (or Displacement Value) based on automatically calculated maximum intensities or based on user-selected best focus slices. Start and End positions can be any period where the plate surface/biofilm can be visualized, thus the biofilm doesn't need to be present in each field of view for the whole experiment. The manual slide selection method is also compatible with brightfield images, and it is usually the best option when fluorescence is very weak. The automatic method is based on fluorescence intensity and is designed to set the central Z-position at the point with the highest intensity. Again, the user can choose specific reference time points for this calculation and the macro will extrapolate to allow processing of the whole image stack.

The second processing phase (Fig. S3) is performed using the calculated the displacement value to apply a floating Z-window crop progressively over time and follow the plate surface position with a user-specified thickness in slices. The software will try to estimate the starting crop position based on manual or automatic values, but users can also manually specify both starting position and displacement value. During processing, the macro writes time points as temporary files (.tif) to avoid memory issues, before reassembling the reduced data set with its original LUT (color) values and calibration. The benefits of BacLive can be seen in Fig. 2D, where a 27.4 Gb data set can be quickly reduced to a much more manageable 3.4 Gb one, without losing the 3D interaction dynamics of the two colonies. The data processed generates a tif file that can be later used in FIJI and other tools such as Imaris, BiofilmQ, or Comstat (19, 28) for

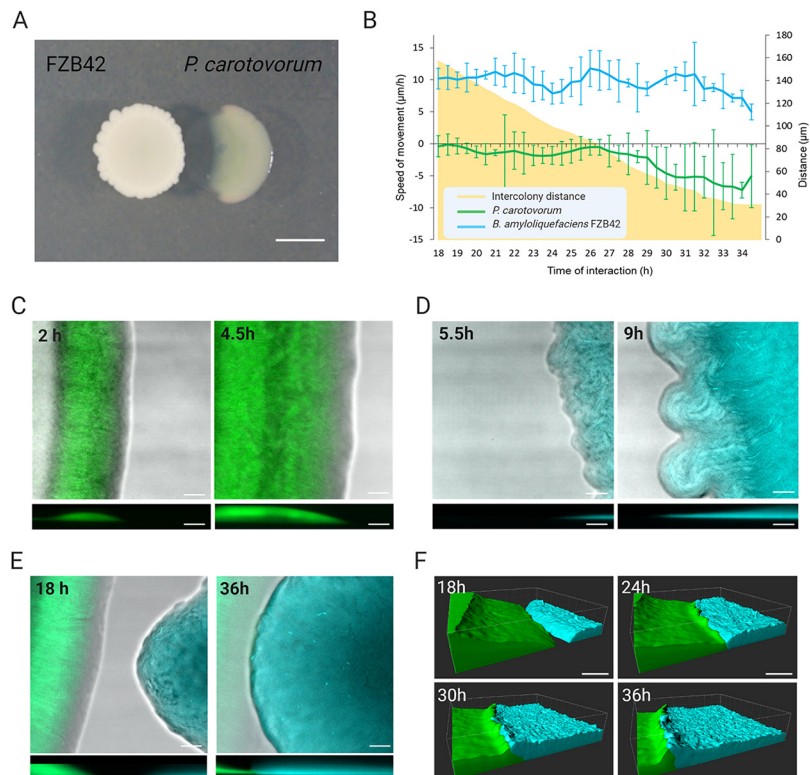

**FIG 3** FZB42 inhibits the initial growth of *P. carotovorum* and they grow in a wall-like structure when getting in contact. (A) Pairwise interaction between FZB42 and *P. carotovorum* shows inhibition of *P. carotovorum* colony from a macroscopic perspective. Scale = 1 cm. (B) Expansion rates of the FZB42 and *P. carotovorum* leading edges and distance between both strains in the long-term stage of the interaction (18 to 36 h). The blue line represents the FZB42 leading edge, and the green line represents the *P. carotovorum* leading edge. The yellow area represents the distance between the two populations during this stage of the interaction. Error bars indicate SD. *n* = 3. (C) 2D images and Z-projected representations of the *P. carotovorum* growth in the short-term time frame of interaction (2 and 4 to 5 h). (D) 2D images and Z-projected representations of FZB42 growth in the short-term time frame of interaction (5.5 and 9 h). (E) 2D images and Z-projected representations of the interaction area where *P. carotovorum* and FZB42 come into contact (18 and 36 h). (F) 3D surface representations of the interaction area show the initial contact between *P. carotovorum* and FZB42 and the FZB42 advancement. In addition, both colonies form a wall-like structure when they come into contact. Scale = 50 $\mu$m.

data analysis and 3D generation of bacterial colonies, just simplifying the preprocessing needed for the use of the mentioned tools.

**(iii) Study of bacterial population dynamics in interactions on solid media.** To illustrate how the BacLive imaging and processing methodology (resumed in Fig. 1A) can be used for the study of practical problems in biofilm dynamics and bacterial interactions, we focused on two examples: (i) bacterial interactions between *Pectobacterium* and *Bacillus* species, and (ii) the expression pattern and spatial distribution of different cell types in a single *Bacillus subtilis* biofilm.

The first example focuses on interactions between two ubiquitous soil species populations: *Pectobacterium carotovorum*, a phytopathogenic species that produces and secretes cell wall degrading enzymes leading to rotting and decay of their plant hosts (29), and *Bacillus amyloliquefaciens* FZB42 (referred as FZB42), a plant beneficial bacterium with antibacterial and antifungal activity against a wide range of plant pathogens (30). Initially, pairwise interactions on LB plates for 48 h between *P. carotovorum* and FZB42 showed the inhibition of *P. carotovorum* growth in the interaction area, where an inhibition halo was formed (Fig. 3A). To study this effect in more detail, we analyzed interaction dynamics using the BacLive method. The FZB42 strain was fluorescently labeled with CFP, while *P. carotovorum* was labeled with GFP, as described in the

Materials and Methods, and spotted on LB solid media at 0.7 mm on glass-bottomed petri dishes. The temperature was controlled and maintained at a constant 28°C during the experiment. Several Z-stacks at 6 different positions were recorded over 24 to 48 h to analyze the leading edge of each colony and the predicted zones of physical contact between the colonies (Fig. S4A). After data acquisition and image processing using the BacLive macro, we could analyze and quantify colony interactions at the different positions monitored. Initially, both bacterial colonies were growing separately and advanced at constant rates (Fig. 3B to D). Time-lapse images at two positions (Fig. 3C and D) allowed us to calculate the expansion rate of *P. carotovorum* at 40 $\mu$m/h and FZB42 at 100 $\mu$m/h after 8 h of growth (Fig. S4B). Both colonies reached one of the intermediate acquisition zones after 15 h of growth. Here, we observed the arrest of *P. carotovorum* colony growth while FZB42 continued its movement toward *P. carotovorum* (Fig. 3E and Video S2). Measurement of colony movement confirmed this observation: FZB42 advanced at a speed of 10 $\mu$m/h, while *P. carotovorum* had initially stopped its advance, and then retreated at a rate of 5 $\mu$m/h (Fig. 3B). The data collected allowed us to measure the distance between the two leading edges, showing that, at the moment the two colonies were recorded in the intermediate area, they were 160 $\mu$m apart and this distance progressively diminished until the colonies were practically in contact (Fig. 3B). These results are surprising because the macroscopic view of the interaction (Fig. 3A) showed a clear inhibition of *P. carotovorum*. However, our microscopic time-lapse data indicate that a viable *P. carotovorum* population is still present in the leading edge of the colony and this results in cell-to-cell contact with FZB42. To get a deeper understanding of the interaction, we took advantage of the 3D Baclive data set and analyzed the development of the two populations using Imaris software. From 18 h and 36 h of interaction, we could observe the death of *P. carotovorum* at the time of continuous FZB42 advancement. Interestingly, when the two colonies came into contact, their leading edges became thicker (Fig. 3F and Video S3 and S4). This suggests specific short-range interactions between the two populations. By combining BacLive with mutational studies, in the future, it may be possible to understand the molecular basis of these interactions. Our results show how this noninvasive technique combining fluorescence labeling with long-term three-dimensional acquisition can provide unique insights into the complex interactions between different bacterial populations.

**(iv) Analysis of bacterial subpopulations during the *Bacillus subtilis* biofilm formation.** Several studies have described the division of labor occurring in a bacterial colony during biofilm formation (21, 31). However, as mentioned above, the use of imaging technology for the study of subpopulations has typically required invasive sample processing, which typically risks deformation of colony morphology, and limits the study to specific time points (20). With the use of the BacLive methodology described in the manuscript, we can follow the development of biofilm formation and the initiation of bacterial subpopulations and their development at the cellular level. To exemplify this, we used a *B. subtilis* biofilm as a model to evaluate the timing and spatial distribution of two subpopulations: (i) cells expressing TasA, a protein involved in the correct formation of the extracellular matrix (15, 32, 33); and (ii) colony expansion based on *motA* expression, required for flagellar motor rotation. Promoters of both genes were placed upstream of sequences encoding two different fluorescent proteins (P*tasA*-mcherry and P*motA*-YFP) and integrated into two different neutral loci, *lacA,* and *amyE*, respectively. 0.7 $\mu$L of the resuspended bacterial solution was used to inoculate Msgg solid media and the experiment was performed as described in the previous section (see the Materials and Methods). We recorded a total of 6 different plate positions to analyze changes at the colony border and more central parts of the colony (Fig. 4A). In addition, two extra recording positions were placed to capture later stages of *B. subtilis* colony expansion. Images were initially processed with BacLive, and then with Fiji and Imaris for further processing, visualization, and analysis. When analyzed as a 2D projection, we could obtain a preliminary idea about the timing of

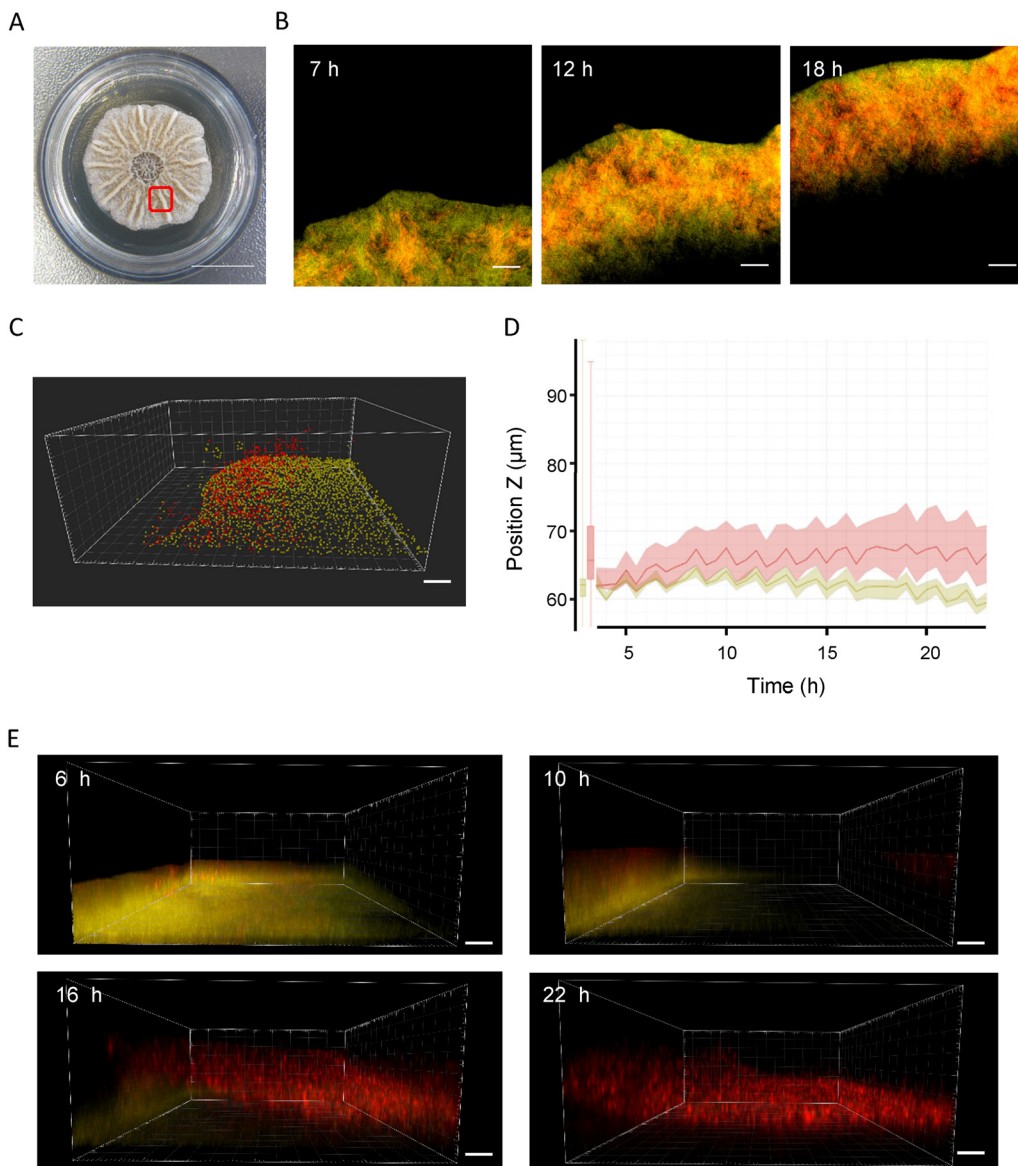

**FIG 4** TasA expressing subpopulation grows in the upper region of the biofilm while motility cells expand *Bacillus* colony in the contact with the culture media. (A) *Bacillus* colony grown in Msgg culture media exhibiting its typical biofilm growth. A red square indicates one of the regions analyzed with the presence of wrinkles. Scale = 1 cm. (B) 2D images representing both subpopulations during the colony growth (TasA expressing cells are fluorescently labeled with mCherry (red) while motility cells expressing MotA are labeled with YFP (yellow). Scale = 20 $\mu$m. Deviation from the mean is highlighted in clear red and yellow and indicates SD. $n$ = 3. (C) 3D image of a *Bacillus* colony growing on Msgg. Each spot represents cells expressing TasA (red) and MotA (yellow) which can be later used for quantification of bacterial subpopulations and localization. Scale = 20 $\mu$m. (D) Z position of TasA and MotA expressing cells over time. Cells expressing TasA (red) are in the upper region of the colony while MotA expressing cells (yellow) are located in the bottom region. (E) 3D representation of the colony growth expressing TasA and MotA. Four snapshots of the colony growth development is shown at different time points (6, 10, 16, and 22 h). TasA cells at 16 and 22 h show the formation of the typical wrinkle found in a *B. subtilis* biofilm. Scale = 20 $\mu$m.

promoter expression. However, following the exact localization of motile cells and TasA producers was not easy (Fig. 4B and Video S5). As expected, promoter expression in the inner colony gradually disappeared due to microbiological factors, such as the bacterial cell death in old areas of the colony, the specific expression of certain promoters depending on bacterial growth, or the loss of the activity of fluorescent proteins at long time points. We next analyzed biofilm formation in 3D using IMARIS, which allowed us to evaluate the relative position of the two bacterial subpopulations with respect to the agar surface. For that, using IMARIS, we identified fluorescent

bacteria foci as spots and measured their distance from the agar surface over time using the 3D BacLive-processed data at each time point (Fig. 4C and Video S6). Fig. 4D shows the average Z-position (height) of the two subpopulations with respect to the agar surface over time. The results clearly show that cells expressing TasA are mostly located in the upper region of the colony involved in extracellular matrix formation while cells expressing MotA are in closer contact with the solid media, thus permitting colony expansion. Significantly, one of the positions selected for time-lapse analysis coincided with a site of wrinkle formation. Wrinkles are structures typical of those formed in a *Bacillus subtilis* biofilm and are known to be empty structures permitting the movement of water and nutrients inside the colony (34). Interestingly, our time-lapse analysis showed the role of motile cells and TasA-producing cells in wrinkle formation (Fig. 4E and Video S6 and S7). While motile cells were associated with initial colonization and the expansion of colony borders, they did not appear to coincide with wrinkle formation. On the other hand, we observed TasA expressing cells adopting a distribution coinciding in time and space with wrinkle formation (Fig. 4E).

**Concluding remarks.** The study of microorganisms at the microscopic level has been pursued since the invention of microscopy in the 17th century. In recent years, the study of bacterial populations and bacterial structures such as biofilms have gained interest and importance. A variety of strategies have been developed to follow these processes over time. At the microscopic level, most existing methods involved a sample processing step before imaging, making it impossible to perform a continuous study over time, and increasing the probability of introducing artifacts into the results.

Recently, some novel strategies have been introduced to improve our ability to image these processes. Nadezhdin et al. (35), for example, have developed a new method to study living biofilms using agar segments, thus facilitating bacterial growth and time-lapse microscopy across a cross-section of the biofilm. This approach has allowed the identification of spatial patterns of gene expression focused on the top of the live biofilm. However, the 2D images obtained may be insufficient for the analysis of larger biofilm regions (Table 1).

In the manuscript, we described a new strategy, BacLive, that provides a new method for studying 3D colony-level interactions and living biofilms without sample manipulation. The use of high-quality long-working distance objective lenses, and glass-bottomed petri dishes in combination with fluorescently labeled bacteria permit the capture of substantial amounts of data that can be conveniently processed using the BacLive macro. This macro greatly simplifies the amount of data generated during the image acquisition, permitting easier data analysis. The combination of this strategy with processing tools such as Imaris, BiofilmQ, or Comstat, has the potential to greatly aid the study of genetic processes in living bacteria, and in obtaining a more complete perspective of the types of interactions that occur in the environment, as well as be valuable in applied sciences, for example in the search and development for antimicrobials.

## MATERIALS AND METHODS

**Strains, media, and culture conditions.** Routinely, bacterial cells were grown in liquid Lysogeny Broth (LB: 1% tryptone (Oxoid), 0.5% yeast extract (Oxoid), and 0.5% NaCl) medium at 28°C (*P. carotovorum*, *B. amyloliquefaciens* FZB42, and *B. subtilis*) with shaking on an orbital platform. Pairwise interaction experiments were performed on LB media while biofilm assays were done on MSgg medium: 100 mM morpholinepropane sulfonic acid (MOPS) (pH 7), 0.5% glycerol, 0.5% glutamate, 5 mM potassium phosphate (pH 7), 50 $\mu$g/mL tryptophan, 50 $\mu$g/mL phenylalanine, 50 $\mu$g/mL threonine, 2 mM MgCl$_2$, 700 $\mu$M CaCl$_2$, 50 $\mu$M FeCl$_3$, 50 $\mu$M MnCl$_2$, 2 $\mu$M thiamine, 1 $\mu$M ZnCl$_2$. *Bacillus subtilis* 168 is a domesticated strain used to transform the different constructs into *Bacillus subtilis* NCIB3610. When necessary, antibiotics were added to the media at appropriate concentrations. Strains and plasmids were constructed using standard methods (36).

**Construction of fluorescence-labeled strains.** For the construction of the double-fluorescently labeled *B. subtilis* strain, P$_{tasA}$ was amplified using primers containing EcoRI and XbaI restriction sites and cloned into plasmid ECE756 previously digested with the same restriction enzymes. The resulting fusion was amplified and cloned into the pDR183 plasmid. P$_{motA}$ was inserted into a pKM003 plasmid. The resulting plasmids were transformed by natural competence into *B. subtilis* 168 replacing the *lacA* and *amyE* neutral locus, respectively. All the *B. subtilis* strains generated were constructed by transforming *B. subtilis* 168 via its natural competence and then using the positive clones as donors for transferring the

constructs into *B. subtilis* NCIB3610 via generalized SPP1 phage transduction (37). Fluorescence labeling plasmid pKM008V was constructed for *B. amyloliquefaciens* FZB42. For that, the $P_{veg}$ promoter fragment (300 bp) was extracted from pBS1C3 by digestion with EcoRI and HindIII restriction enzymes, purified, and cloned in this case into a pKM003 plasmid, which was previously digested with the same restriction enzymes. pKM008V was then linearized and transformed into *B. amyloliquefaciens* FZB42 by natural competence and transformants selected by plating on LB agar plates supplemented with spectinomycin (100 $\mu$g/mL). *P. carotovorum* was fluorescently labeled by electroporation using the pRL662-gfp kindly donated by the Ehr-Min Lai laboratory.

**Time-lapse microscopy.** Bacterial interactions and biofilm dynamics in the solid medium were visualized by confocal laser scanning microscopy (CLSM). For biofilm studies, 0.7 $\mu$L of the double-labeled *B. subtilis* strain were spotted on MSgg solid media while for bacterial interaction time-lapse experiments, *B. amyloliquefaciens* FZB42 and *P. carotovorum* labeled strains were spotted at a 0.5 cm distance onto 1.3 mm thick LB agar plates using 35 mm glass-bottomed dishes suitable for confocal microscopy (glass-bottom dish 35 mm, Ibidi, catalog number 81218-200). Plates were incubated at 28°C for 6 h before acquisition. The temperature was maintained at 28°C during the time course using the integrated microscope incubator. Acquisitions were performed using an inverted Leica SP5 confocal microscope with an HC FLUOTAR L 25×/0.95 W VISIR long working distance water immersion objective. Bacterial fluorescence could be visualized from underneath the bottom of the plate and through the agar medium thanks to the long 2.2 mm free working distance of this objective. A special oil immersion medium, Immersol W 2010 (Carl Zeiss Immersion Oil Immersol Fisher scientific, product code: 11825153) was used instead of water to avoid problems with evaporation during the experiment. Because oil does not have the same surface tension as water, there is a limit to how much oil can be added without it dripping down from the lens-dish interface. Fortunately, because the objective is relatively close to the base of the petri dish (i.e., at or near to the limit of its focal distance), only a relatively small amount of immersion medium is required. Colony fluorescence was followed in multiple regions selected at the start of the experiment, with the acquisition of a series of different focal (z) positions at each region performed automatically at every time point. Evaporation from the LB agar and its utilization by the growing colonies results in a gradual lowering of the agar surface relative to the objective lens of approximately 250 $\mu$m every 24 h. To be able to follow colony dynamics, images were acquired over a wide focal range to compensate for the predicted change in colony position during the experiment. Image processing and 3D visualization were performed using ImageJ/FIJI[79,80], and Imaris version 7.6 (Bitplane). Expansion rates and distance between colonies were calculated with FIJI, using the change in position of the leading edge of the colony between time points. Expansion rate results were calculated as an average speed at three different regions of the same colony, with variations expressed as standard deviation while distance was calculated between the closest points of both leading edges of the colonies. To reduce the impact of random vibrations and variations, plotting speed values were smoothed as a floating 4-value average advancing 30 min (or 1 timepoint) at a time. Experiments were done in triplicate and results are represented as means ± standard deviation (SD).

## SUPPLEMENTAL MATERIAL

Supplemental material is available online only.

**SUPPLEMENTAL FILE 1**, PDF file, 0.5 MB.

**SUPPLEMENTAL FILE 2**, AVI file, 2.3 MB.

**SUPPLEMENTAL FILE 3**, AVI file, 16.6 MB.

**SUPPLEMENTAL FILE 4**, AVI file, 15.9 MB.

**SUPPLEMENTAL FILE 5**, MP4 file, 12 MB.

**SUPPLEMENTAL FILE 6**, MP4 file, 18 MB.

**SUPPLEMENTAL FILE 7**, MP4 file, 11.6 MB.

## ACKNOWLEDGMENTS

We thank Saray Morales Rojas for technical support.

This work was supported by grants from ERC Starting Grant (BacBio 637971) and Plan Nacional de I+D+i of Ministerio de Economía y Competitividad and Ministerio de Ciencia e Innovación (AGL2016-78662-R and PID2019-107724GB-I00). C.M.-S. is funded by the program Juan de la Cierva Incorporación (IJC2018-036923-I). M.V.B.-C. and A.I.P.-L. are funded by the program FPU (FPU17/03874 and FPU19/00289, respectively).

We declare no conflicts of interest.

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
