## [Reviewer comments · Microbiology Spectrum]

Microbiology Spectrum

A non-invasive method for time-lapse imaging of microbial interactions and colony dynamics

Carlos Molina-Santiago, John Pearson, María Victoria Berlanga-Clavero, Alicia Pérez-Lorente, Antonio de Vicente Moreno, and Diego Romero

Corresponding Author(s): Diego Romero, Universidad de Málaga

Review Timeline:

Submission Date:	March 13, 2022
Editorial Decision:	April 12, 2022
Revision Received:	May 20, 2022
Accepted:	June 10, 2022

Editor: Olaya Rendueles Garcia

Reviewer(s): The reviewers have opted to remain anonymous.

Transaction Report:

DOI: <https://doi.org/10.1128/spectrum.00939-22>

April 12, 2022

Dr. Diego Romero
Universidad de Málaga
Microbiology
Campus de Teatinos
Málaga
Spain

Re: Spectrum00939-22 (A non-invasive method for time-lapse imaging of microbial interactions and biofilm dynamics)

Dear Dr. Diego Romero:

Thank you for submitting your manuscript to Microbiology Spectrum.

Your manuscript has been reviewed by three different experts in the field. I hope you will find their comments useful and fair. They find BacLive could be interesting contribution to the field, but they require some modifications and raise some concerns. Specifically, two reviewers highlight that it is not clear how BacLive would advance our knowledge and understanding of microbial interactions.

Also there are some issues concerning repeatability. First, I believe it is important you provide more technical details (as raised by reviewer #1), as well as raw data files in a repository for easy testing. It would be useful to compare your tool to others established in the field, as suggested by reviewer #2. Finally, reviewer #3 raises concerns about the quality of the imaging and lack of certain controls.

Link Not Available

Sincerely,

Olaya Rendueles Garcia

Journals Department
Reviewer comments:

Reviewer #1 (Comments for the Author):

In this manuscript the authors showcase BacLive as a novel non-invasive method for tracking bacterial growth in biofilms. I believe the manuscript is timely and will aid researchers who are increasingly looking to utilize imaging and analytic tools to study biofilms. In addition, the use of this tool will also be useful for researchers working on anti-microbial assays, especially in the context of their discovery that macroscopic views of zones of inhibition can be misleading. I have a few suggestions to further improve the text.

Suggestions:

Line 42-43: Suggested rephrasing: from `sample manipulation what deprives from microscopic time lapse experiments` to `sample manipulation that is unamenable to microscopic time lapse experiments`

Line 145: The authors should elaborate why CLSM systems are insensitive compared to widefield setups

Line 149: Again, the authors should explain by what they mean when they use the term sensitivity

Line 176-179: To make their methods easily reproducible by other groups, the authors should state the manufacturer name or incorporate manufacturer codes, product numbers for things like cover slips or Petri Dishes. This can be either in the methods or in the main text.

Line 466: Replace evolution with development. Also, throughout the manuscript in general. Avoid the use of evolution as what you are looking can be best explained as development of biofilms. Also, line 749

Line 483: Explain what the technical and microbiological factors are. Will be useful for planning experiments.

Reviewer #2 (Comments for the Author):

The authors describe a novel analyzing platform BacLive to study the bacterial interactions on agar surfaces. The authors describe the methods and other issues faced during imaging clearly. This paper is, therefore, very useful to biofilm researchers. However, I have a few suggestions to further improve the reader experience and its contribution to the research community.

1. Authors in the introduction state the merits and the shortcomings of the different image acquiring and analyzing methods used currently to study the bacterial interactions on the agar surfaces using microscopic techniques. However, they fail to cite some of the prominent studies in the field addressing this. For example, work from Suel, Foster, and Drescher labs.
2. In Results, the acquisition methodology is written like a review. It would be ideal if authors make it clear how they have done things. In the current format, the description is wordy and emphasizes more on how to improve techniques in a general way.
3. Please comment on how the objective lens images through solid agar. MSgg and LB can be really dense to visualize bacteria through the agar surface.
4. Please comment on how BacLive is different from the current processing tools available with ImageJ Bioformats. Can this tool be used with Comstat, BiofilmQ, and other tools, such as Imaris's own biofilm analysis extension?
5. Lastly, the authors state how the tool could be useful to study bacterial interactions. What features make this tool novel?

Reviewer #3 (Comments for the Author):

This paper describes a time-lapse microscopy approach to analyzing interactions between agar-supported microbial colonies. Though this reviewer is a proponent of direct visualization approaches, there are a collection of deficiencies of the model as described.

1. In the title, it would be more accurate to replace "microbial interactions and biofilm dynamics" with "microbial interactions and dynamics in colonies."
2. The imaging is of poor quality. This is likely due to having to image through a 1mm thick layer of agar which greatly compromises resolution.
3. Imaging within the cross-section of colonies (z-stacks) is also poor; I do not believe these constitute effective 3D imaging. Colonies can be quite opaque and excitation and emission intensities can be sharply attenuated even at short distances into the bacterial aggregate. The limited penetration of light into and out of the colony is not measured or discussed.
4. Evaporation causes the agar to dry and shrink, resulting in the vertical movement of colonies throughout the imaging. This system would be much improved by engineering to control humidity.

5. On the page after Table 1, there is a figure that does not appear to have a corresponding figure caption.
6. Repeatability between experiments is not adequately (quantitatively) addressed.
7. The examples shown are not sufficient to provide compelling and quantitative demonstration of microbial interactions. It would be helpful to have multiple controls, for example, for a positive interaction, a negative interaction, no interaction, etc.

Staff Comments:

Preparing Revision Guidelines

Please return the manuscript within 60 days; if you cannot complete the modification within this time period, please contact me. If you do not wish to modify the manuscript and prefer to submit it to another journal, please notify me of your decision immediately so that the manuscript may be formally withdrawn from consideration by Microbiology Spectrum.

Reviewer comments:

Reviewer #1 (Comments for the Author):

Q - In this manuscript the authors showcase BacLive as a novel non-invasive method for tracking bacterial growth in biofilms. I believe the manuscript is timely and will aid researchers who are increasingly looking to utilize imaging and analytic tools to study biofilms. In addition, the use of this tool will also be useful for researchers working on anti-microbial assays, especially in the context of their discovery that macroscopic views of zones of inhibition can be misleading. I have a few suggestion to further improve the text.

A – Thank you very much for your comments and suggestions. We have tried to address the issues highlighted in the new text and believe in the manuscript has been significantly improved as a result. In addition, as suggested by the editor, we have uploaded a test file to be used in BacLive: (<https://www.dropbox.com/s/8vaxqkwd7yjjph/BacLive%20Test.7z?dl=0>). Reviewers can download the file and test BacLive tool (<https://github.com/BacLive>). The same test data link is also shared within the Github repository to help other users get started with this method.

Suggestions:

Q - Line 42-43: Suggested rephrasing: from `sample manipulation what deprives from microscopic time lapse experiments` to `sample manipulation that is unamenable to microscopic time lapse experiments`

A – Thank you, we have modified the sentence.

Q - Line 145: The authors should elaborate why CLSM systems are insensitive compared to widefield setups

A – We have introduced changes to the text (Lines 171-184) to explain the relative sensitivity of confocal and widefield microscopy in more detail, including citations of relevant review articles. Moreover, in response to comments from all three reviewers, we have rearranged the structure of the first part of the results section to give greater prominence to issues specific to the BacLive method as well as improve the general clarity and readability of the text.

Q - Line 149: Again, the authors should explain by what they mean when they use the term sensitivity

A – We have clarified this concept as part of the changes mentioned above (lines 171-184).

Q - Line 176-179: To make their methods easily reproducible by other groups, the authors should state the manufacturer name or incorporate manufacturer codes, product numbers for things like cover slips or Petri Dishes. This can be either in the methods or in the main text.

A – We have included extensive product details (Manufacturer, product name and product codes) in the Materials and Methods section.

Q - Line 466: Replace evolution with development. Also, throughout the manuscript in general. Avoid the use of evolution as what you are looking can be best explained as development of biofilms. Also, line 749

A – Done, thank you.

Q - Line 483: Explain what the technical and microbiological factors are. Will be useful for planning experiments.

A – We have added some technical and microbiological factors (See lines 497-500).

Reviewer #2 (Comments for the Author):

Q - The authors describe a novel analyzing platform BacLive to study the bacterial interactions on agar surfaces. The authors describe the methods and other issues faced during imaging clearly. This paper is, therefore, very useful to biofilm researchers. However, I have a few suggestions to further improve the reader experience and its contribution to the research community.

A – Thank you for the positive feedback. We believe that by incorporating these suggestions the manuscript has been significantly improved.

Q - 1. Authors in the introduction state the merits and the shortcomings of the different image acquiring and analyzing methods used currently to study the bacterial interactions on the agar surfaces using microscopic techniques. However, they fail to cite some of the prominent studies in the field addressing this. For example, work from Suel, Foster, and Drescher labs.

A – The reviewer is correct. We have incorporated examples from Suel, Foster and Drescher labs into the Introduction. See lines 112-114 and 126-128.

Q - 2. In Results, the acquisition methodology is written like a review. It would be ideal if authors make it clear how they have done things. In the current format, the description is wordy and emphasizes more on how to improve techniques in a general way.

A – Thank you. In response to this suggestion, as well as other reviewer comments, we have rearranged the first part of the results section to give more prominence and focus to the BacLive method itself. We believe the revised text is easier to follow and explains our approach much more clearly. In terms of acquisition methodology, we have included additional information explaining our protocol in detail (see lines 190-206).

Q - 3. Please comment on how the objective lens images through solid agar. MSgg and LB can be really dense to visualize bacteria through the agar surface.

A – Although the culture media used are relatively dense, they are almost transparent and homogeneous. Variations in the refractive index within a non-homogenous solution or media is much more important than density in terms of sample opacity, microscope imaging penetration and image quality. This can be readily seen in solutions whose ingredients are not fully dissolved. Indeed, we observed little or no attenuation of fluorescence signal through the ~1 mm of MSgg or LB solid agar. In contrast, fluorescence signal attenuates relatively rapidly as it crosses into bacterial colonies, which are less dense but composed of complex mixtures of proteins, water and lipids, making it much more difficult for fluorescence excitation and emission to pass through it. As this concept is fundamental to the BacLive method, we have added an additional text (Lines 153-167) explaining these ideas more explicitly accompanied with relevant references

and new supplementary figure to illustrate the imaging depth and the imaging quality possible with this method (Suppl. Figure 1).

In addition, we have included our experiments assaying different thickness of the solid agar in order to confirm the best conditions to visualize bacteria through the agar surface (See Suppl. Figure 2) to ensure the optimal combination of solid agar media thickness for growth and imaging for our experimental strains.

Q - 4. Please comment on how BacLive is different from the current processing tools available with ImageJ Bioformats. Can this tool be used with Comstat, BiofilmQ, and other tools, such as Imaris's own biofilm analysis extension?

A – To the best of our knowledge, there are no existing ImageJ plugins able to detect and compensate for constant focus changes in long-term imaging experiments in the way that the BacLive macro does. BacLive is optimized to allow the processing of very large datasets (30gb+) even with systems that do not have large amounts of RAM. On the other hand, BacLive is fully compatible and complimentary to other ImageJ plugins and image processing packages. For example, to import raw data in the Leica .lif format we normally use the Bioformats plugin prior to processing with the BacLive macro. It is also compatible with the virtual stack option offered by Bioformats (required when opening datasets larger than the available RAM). Once processed, the much smaller, stabilized dataset retains its calibration and size metadata allowing for further processing and analysis using other ImageJ plugins and packages including biofilm-specific BiofilmQ (Biofilm focused image processing and analysis package) or Comstat (Matlab-based method for Quantification and statistical analysis of biofilm structures).

We have added a paragraph highlighting the possibility of using BacLive as a first step in the processing of imaging for later use with other programs and plugins (See lines 422-425).

Q - 5. Lastly, the authors state how the tool could be useful to study bacterial interactions. What features make this tool novel?

A – From our point of view, the BacLive method described in this manuscript is novel since it permits the high-resolution 3D visualization of bacterial interactions, biofilms and bacterial colony development in general using a non-invasive methodology. It thus provides a tool to decipher mechanisms involved in different bacterial processes such as biofilm development or the resistance against antimicrobials secreted by cohabitant bacteria. In fact, as shown in Molina-Santiago et al., 2019 (<https://doi.org/10.1038/s41467-019-09944-x>), we have demonstrated that this method is useful to decipher the role of the extracellular matrix components in bacterial interactions. Furthermore, the use of the BacLive macro for the analysis of data is, from our perspective, highly valuable in facilitating the analysis of large datasets using standard laboratory computers, making non-invasive microscopic studies of bacterial interactions more accessible and affordable. We have incorporated information highlighting the novelty and relevance of BacLive along the results and conclusion sections (e.g., lines 153-157).

Reviewer #3 (Comments for the Author):

This paper describes a time-lapse microscopy approach to analyzing interactions between agar-supported microbial colonies. Though this reviewer is a proponent of direct visualization approaches, there are a collection of deficiencies of the model as described.

Q - 1. In the title, it would be more accurate to replace "microbial interactions and biofilm dynamics" with "microbial interactions and dynamics in colonies."

A – We have modified the title, thank you for the suggestion.

Q - 2. The imaging is of poor quality. This is likely due to having to image through a 1mm thick layer of agar which greatly compromises resolution.

A – We thank Referee #3 for their valid observation and we are grateful for the opportunity to clarify this important issue. As we discussed in the original manuscript, long-term timelapse imaging always involves compromises in terms of image quality, temporal resolution, number of imaging positions, illumination intensity etc. The examples provided in this article were focused on long-term (3-5 days) analyzing multiple positions in nascent colonies and intermediate positions where interactions may occur. Thus, to avoid generating prohibitive amounts of data and to maintain a temporal resolution below 30 minutes, we choose an image format of 512 x 512 pixels and zoom levels well below the maximum obtainable resolution for our system but was still sufficient for the goals of those particular experiments. As we describe above in our reply to Reviewer #2, the density of ~1 mm agar substrate does not significantly affect imaging quality or imaging depth due to its homogenous refractive index. In contrast the less dense but more complex mixture of lipids, proteins, nucleic acid and water, with a mixture of different refractive indices found in a bacterial colony is much more challenging from an imaging perspective with significant attenuation occurring within ~50-75 microns. In addition to the new paragraph highlighting the importance of a homogenous refractive index for imaging applications (Lines 153-167), we have added a new information related with the relevance of choice between image quality, data size and temporal resolution depending on the experiment (Lines 309-316 and 340-348) and a new supplementary figure (Suppl. Figure 1) shows examples of BaLive imaging at higher resolutions that are closer to the maximum resolution of the objective lens used. In this way we hope the revised manuscript better explains the strengths and flexibility of the BaLive method and the different trade-offs required for a given experiment.

Q - 3. Imaging within the cross-section of colonies (z-stacks) is also poor; I do not believe these constitute effective 3D imaging. Colonies can be quite opaque and excitation and emission intensities can be sharply attenuated even at short distances into the bacterial aggregate. The limited penetration of light into and out of the colony is not measured or discussed.

A – The axial resolution of the examples presented in our manuscript is limited by the 2-3 μm sampling frequency used in these cases. This represents another compromise to perform these long term, multi-position experiments rather than a limitation of the method *per se*. We completely agree that the attenuation of fluorescence within bacterial colony limits the depth at which useful information can be obtained. However, depth attenuation is common to any non-invasive fluorescence imaging method and does not preclude effective or useful 3D imaging. In Molina-Santiago et al., 2019 (<https://doi.org/10.1038/s41467-019-09944-x>), using the same methodology described in the present manuscript, we effectively tracked global colony interactions in 3D using mCherry and CFP, while being able to track colony invasion by individual mCherry positive bacteria in 3D. In this current manuscript, we also demonstrate the ability to track the 3D position of bacterial foci within a single colony expressing different fluorescent reporter constructs relative to the agar substrate (Figure 4). We have added a discussion of the effects of fluorescence depth attenuation and limitations it brings to the main text (Lines 153-167). The additional Suppl. Figure 1 specifically shows orthogonal cross-sections of a bacterial colony with bacterial fluorescence detectable imaging up to ~75 microns from the substrate surface.

Q - 4. Evaporation causes the agar to dry and shrink, resulting in the vertical movement of colonies throughout the imaging. This system would be much improved by engineering to control humidity.

A – We completely agree with the reviewer. In fact, we discuss this point in the section “Methods for dealing with evaporation and media shrinkage”. It is true that some approaches like the one applied in Peñil-Cobo et al., 2018 could be incorporated into our system to further reduce the agar shrinkage. However, we have found our method to effectively compensate for agar shrinkage as is shown in Figure 2C and the movement can be easily calculated for. Nevertheless, we are looking towards future improvements of the BaLive method that can improve humidity control and reduce evaporation from the solid media. We have specified the approach used by Peñil-Cobo et al. in this section of the manuscript as a potential improvement of our method (Lines 254-257).

Q - 5. On the page after Table 1, there is a figure that does not appear to have a corresponding figure caption.

A – The reviewer is right. This was a mistake during the uploading of the documents and figures. Figure after Table 1 was, indeed, Figure 4. We have reorganized the files and uploaded correctly in this version of the manuscript.

Q - 6. Repeatability between experiments is not adequately (quantitatively) addressed.

A – Experiments were done at least in triplicates and measurement of expansion rates and distance were calculated as a mean of the replicates. We have now clarified this information in the figure legends and we have incorporated standard deviation to all the plots shown in the manuscript. See legends of Figures 3, 4 and Suppl. Figure 4.

Q - 7. The examples shown are not sufficient to provide compelling and quantitative demonstration of microbial interactions. It would be helpful to have multiple controls, for example, for a positive interaction, a negative interaction, no interaction, etc.

A – We believe that the examples shown in this methods article clearly demonstrate the value of BaLive method for following biofilm dynamics, colony interactions and cellular differentiation in different colony regions using both comparative and quantitative approaches. Moreover, we have previously demonstrated how this method can be used to compare and quantify changes to colony interactions in wildtype control and mutant conditions (Molina-Santiago et al., 2019, <https://doi.org/10.1038/s41467-019-09944-x>). We hope that the improvements to the new manuscript based on reviewer suggestions, including examples of different imaging configurations, how it compliments other biofilm-analysis packages and the modified graphs showing standard deviation, are now sufficient to demonstrate the value of this flexible methodology for quantitative studies of biofilms and microbial interactions.

June 6, 2022

Dr. Diego Romero
Universidad de Málaga
Microbiology
Campus de Teatinos
Málaga
Spain

Re: Spectrum00939-22R1 (A non-invasive method for time-lapse imaging of microbial interactions and colony dynamics)

Dear Dr. Romero, dear Diego:

Your manuscript has been accepted, and I am forwarding it to the ASM Journals Department for publication. You will be notified when your proofs are ready to be viewed.

One last suggestion has been mentioned by a reviewer. This reviewer believes it would be best to make it clear that the software is useful to study the colony-level interactions on the agar surface and not bacterial interactions entirely (that includes single-cell interaction).

In order to no further delay the publication process, I would urge you to make this small edit once you received the proofs.

Sincerely,

Olaya Rendueles Garcia
Editor, Microbiology Spectrum

Journals Department
Supplemental file 7: Accept
Supplemental file 6: Accept
Supplemental file 5: Accept
Supplemental Material: Accept
Supplemental file 10: Accept
Supplemental file 9: Accept
Supplemental file 8: Accept